# COVID-19 Causes Ferroptosis and Oxidative Stress in Human Endothelial Cells

**DOI:** 10.3390/antiox12020326

**Published:** 2023-01-31

**Authors:** Stanislovas S. Jankauskas, Urna Kansakar, Celestino Sardu, Fahimeh Varzideh, Roberta Avvisato, Xujun Wang, Alessandro Matarese, Raffaele Marfella, Marcello Ziosi, Jessica Gambardella, Gaetano Santulli

**Affiliations:** 1Department of Medicine, Division of Cardiology, Wilf Family Cardiovascular Research Institute, Einstein Institute for Aging Research, Albert Einstein College of Medicine, New York, NY 10461, USA; 2University of Campania “Luigi Vanvitelli”, 81100 Caserta, Italy; 3“Federico II” University, 80131 Naples, Italy; 4Cardarelli Hospital, 80131 Naples, Italy; 5New York Genome Center, New York, NY 10013, USA; 6Department of Molecular Pharmacology, Einstein Institute for Neuroimmunology and Inflammation (INI), Einstein-Mount Sinai Diabetes Research Center (ES-DRC), Fleischer Institute for Diabetes and Metabolism (FIDAM), Albert Einstein College of Medicine, New York, NY 10461, USA

**Keywords:** COVID-19, endothelial dysfunction, ferroptosis, HUVEC, inflammation, lipid peroxidation, long COVID, oxidative stress, oxytosis, peroxidation, ROS, SARS-CoV-2

## Abstract

Oxidative stress and endothelial dysfunction have been shown to play crucial roles in the pathophysiology of COVID-19 (coronavirus disease 2019). On these grounds, we sought to investigate the impact of COVID-19 on lipid peroxidation and ferroptosis in human endothelial cells. We hypothesized that oxidative stress and lipid peroxidation induced by COVID-19 in endothelial cells could be linked to the disease outcome. Thus, we collected serum from COVID-19 patients on hospital admission, and we incubated these sera with human endothelial cells, comparing the effects on the generation of reactive oxygen species (ROS) and lipid peroxidation between patients who survived and patients who did not survive. We found that the serum from non-survivors significantly increased lipid peroxidation. Moreover, serum from non-survivors markedly regulated the expression levels of the main markers of ferroptosis, including GPX4, SLC7A11, FTH1, and SAT1, a response that was rescued by silencing TNFR1 on endothelial cells. Taken together, our data indicate that serum from patients who did not survive COVID-19 triggers lipid peroxidation in human endothelial cells.

## 1. Introduction

We and others have evidenced the fundamental role of endothelial cells in the pathobiology of the coronavirus disease 2019 (COVID-19) caused by the severe acute respiratory syndrome coronavirus (SARS-CoV-2) [1,2,3,4,5,6,7,8,9,10,11,12,13,14,15,16,17]. Equally important, oxidative stress has been shown to be a key player in COVID-19 [18,19,20,21,22,23,24,25]. Chronic inflammation and oxidative stress are associated with endothelial dysfunction, and have been linked to the pathogenesis of atherosclerosis, hypertension, and other cardiovascular and cerebrovascular disorders [26,27,28,29,30,31,32,33,34,35,36,37,38,39,40,41,42,43,44,45,46,47,48].

The overproduction of reactive oxygen species (ROS) is known to induce endothelial dysfunction through different mechanisms, including ferroptosis [49,50], a non-apoptotic type of programmed cell death characterized by the accumulation of lipid peroxides [51,52,53,54,55]. However, to the best of our knowledge, the association of endothelial dysfunction with ferroptosis has never been investigated in the context of COVID-19.

We hypothesized that oxidative stress and subsequent lipid peroxidation induced by COVID-19 in endothelial cells could be linked with the disease outcome. To test this hypothesis, we collected serum from COVID-19 patients on hospital admission, and we incubated these sera with human endothelial cells, comparing the effects on ROS generation and lipid peroxidation between patients who survived and patients who did not survive.

## 2. Results

### 2.1. COVID-19 Induces Oxidative Stress in Human Endothelial Cells

We evaluated the effects on ROS production in human umbilical endothelial cells (HUVECs) of serum obtained from patients who did not survive COVID-19, comparing these effects with the serum of patients who did survive. The main characteristics of the patients are shown in Appendix A. We observed a significantly (*p* < 0.0001) increased production of both cellular (Figure 1) and mitochondrial (Figure 2) ROS in HUVECs incubated for 24 h with 10% serum from non-survivors. 

### 2.2. Serum from COVID-19 Non-Survivors Causes Lipid Peroxidation

Oxidative stress has been linked to lipid peroxidation, a process that has been recently suggested to play a critical role in COVID-19 [56]; on these grounds, we assessed lipid oxidation in endothelial cells, measuring malondialdehyde (MDA), 4-hydroxynonenal (4-HNE), and C11-BODIPY [57,58,59]. We found that all these markers were significantly increased in HUVECs incubated for 24 h with 10% serum from non-survivors compared to serum from COVID-19 survivors (Figure 3), whereas there were no significant differences between normal serum (from healthy donors) and serum from COVID-19 survivors (Figure 3).

### 2.3. Serum from COVID-19 Non-Survivors Reduces the Expression of the Antioxidant Enzyme Glutathione Peroxidase 4 (GPX4) in HUVECs

Since the selenoprotein glutathione peroxidase 4 (GPX4) has been shown to be a major modulator of lipid peroxidation [60,61,62], we reasoned that COVID-19 serum could regulate its expression. Thus, we measured GPX4 in HUVECs, both at the mRNA and at the protein level, and we found that a 24 h incubation with serum obtained from patients who did not survive COVID-19 significantly decreased GPX4 expression (Figure 4).

### 2.4. Serum from COVID-19 Non-Survivors Triggers Ferroptosis in Human Endothelial Cells

Increased lipid peroxidation and reduced levels of GPX4 represent two hallmarks of ferroptosis [63,64]; thus, to verify whether this process was actually involved in our model, we measured the levels of other markers of ferroptosis and their response to Ferrostatin-1 (Fer-1), a well-established inhibitor of ferroptosis [64,65,66]. We observed that in addition to GPX4, suppressors of ferroptosis, such as Solute Carrier Family 7 Member 11 (SLC7A11, also known as Calcium Channel Blocker Resistance Protein, CCBR1, and Cystine/Glutamate Transporter, a subunit of the Amino Acid Transport System X_c_^−^) [67,68,69,70] and Ferritin heavy chain (FTH1) [71,72,73], were significantly downregulated, whereas the ferroptosis inducer spermidine/spermine N^1^-acetyltransferase 1 (SAT1) [74,75,76] was upregulated, in HUVECs incubated with serum from COVID-19 non-survivors; strikingly, these regulations were all prevented by Fer-1 (Figure 5).

### 2.5. COVID-19-Induced Ferroptosis in HUVECs Is Mediated by TNFα

Several studies have consistently shown that increased TNFα serum levels on hospital admission are associated with the risk of mortality in COVID-19 patients [77,78,79]. Similarly, increased serum levels of soluble TNFα Receptor 1 (TNFR1) are associated with mortality [80]. A recent meta-analysis [81] has confirmed that TNFα significantly augments the risk of COVID-19 mortality. Therefore, we measured TNFα in our samples, observing significantly elevated levels in serum from non-survivors compared to serum from survivors (10.7 ± 3.3 * vs. 6.3 ± 4.1 pg/mL, *p* < 0.001). 

Then, to mechanistically prove that TNFα was involved in the COVID-19 induced ferroptotic response, we measured by immunoblot GPX4, SLC7A11, FTH1, and SAT1 after having treated HUVECs with a siRNA for TNFR1 or scramble (the efficiency of TNFR1 silencing is shown in Appendix A). Strikingly, we observed that knocking-down TNFR1 (200 nM siRNA) rescued the regulation of all the ferroptosis markers induced by incubation with serum obtained from patients who did not survive COVID-19 (Figure 5A–E).

## 3. Discussion

The essential contribution of endothelial dysfunction to the pathobiology of COVID-19 was proposed at the beginning of the pandemic in an attempt to explain the systemic manifestations of the disease, and was later verified by investigations on autoptic samples [1,3,4,5,6,7,8,82,83,84,85]. Recent analyses using single-cell atlases have confirmed that endothelial cells in different organs are indeed deeply affected by COVID-19 [86], although the exact mechanisms are still not fully understood.

The association between oxidative stress and programmed cell death, known as oxytosis [87], had been described in neuronal cells years before the term ferroptosis had been coined by Scott Dixon, Brent Stockwell, and collaborators in 2012 [88] to describe a form of non-apoptotic cell death induced by the small molecule erastin by inhibiting SLC7A11. SLC7A11 is a member of a heteromeric, Na^+^-independent, anionic amino acid transport system that is highly specific for glutamate and cysteine [89,90]; in such a system, designated X_c_^−^, the anionic form of cysteine is transported in exchange for glutamate [91,92,93,94,95]. When inhibited, the transportation into the cytosol of cystine, a precursor of glutathione, in exchange for glutamate, is blocked, leading to an impaired function of the housekeeping enzyme GPX4 [96]. The subsequent accumulation of lipid peroxides eventually compromises the plasma membrane integrity [97,98]. It is notable that previous studies exploring cellular metabolism had described similar phenomena, although less detailed from a molecular point of view, in human cells [99]. 

In our experiments, we were able to demonstrate that the main actors involved in the ferroptotic process, namely, GPX4, SLC7A11, FTH1, and SAT1, were all regulated by serum from COVID-19 non-survivors and, in the opposite direction, by TNFR1 silencing, strongly suggesting that TNFα is functionally involved in the association between COVID-19 and ferroptosis in endothelial cells. Although TNFα has been shown in several studies to be a reliable predictor of mortality in COVID-19 [77,81,100,101,102,103], we reckon that other mechanisms could be at play, including the main component of the so-called cytokine storm [103,104,105,106,107,108,109,110,111]. Our observation that not only cellular oxidative stress but also mitochondrial ROS are increased after incubation of HUVECs with COVID-19 serum for non-survivors is consistent with previous reports, which indicate that mitochondria are not simple spectators but active participants in oxytosis and ferroptosis [87,112,113,114,115,116].

In the last years, lipid peroxidation has been implied in a series of disorders, including cancer [117,118,119], neurodegeneration [120,121,122], autoimmune diseases [123,124,125,126], diabetes [127,128,129,130], and ischemia-reperfusion injuries [131,132,133]. With COVID-19, we add another small piece to this jigsaw puzzle, which is in agreement with the functional contribution of ferroptosis to a complex process, such as fibrosis of the lung [134,135,136]. Intriguingly, our results are consistent with the autoptic observations of accumulated oxidized phospholipids in both renal and myocardial tissue in a patient who died because of COVID-19 [137], further indicating that lipid peroxidation and ferroptosis could be playing major roles in the pathophysiology of COVID-19.

Our study is not exempt from limitations, including having performed the assays only in vitro, and not having explored in detail the exact signaling pathway linking TNFα and ferroptosis. In this sense, further dedicated experiments, including in vivo assays, are warranted to corroborate our findings. We did not measure another marker of ferroptosis, heme-oxygenase 1 (HO-1), which has been recently reported both as a suppressor and as an inductor of ferroptosis in a series of conflicting data [138,139].

## 4. Materials and Methods

### 4.1. Cell Culture and Reagents

All reagents were purchased from Merck (Darmstast, Germany), unless otherwise stated. HUVECs were obtained from ThermoFisher Scientific (Waltham, MA, USA; Catalog number: #C0035C). Cells were cultured in a standard humidified atmosphere (37 °C) containing 5% CO_2_, in EGM-2 medium (Lonza, Basel, Switzerland; Catalog number: #CC4147), as we described [12,13,47,140]. Experiments on HUVECs were performed at passages 3–7. HUVECs were plated on glass-bottom culture dishes (MatTek Corporation, Mashland, MA, USA; Catalog number: #P35GCOL-0-10-C). When 70–80% confluent, the cells were treated with 10% human serum for 24 h under normal conditions (37 °C and 5% CO_2_).

The sera from fully de-identified COVID-19 patients were provided by the Montefiore–Einstein COVID-19 biorepository; the protocol was approved by the Institutional Ethical Committee (IRB *#202011756*). Normal human serum was purchased from ThermoFisher Scientific. The absence of SARS-CoV-2 in the serum was confirmed by RT-qPCR. TNFα was quantified via ELISA (ThermoFisher Scientific; Catalog number #KHC3011) following the manufacturer’s protocol.

### 4.2. Assays Measuring Reactive Oxygen Species (ROS) and Lipid Oxidation

ROS production was quantified using 2′-7′-dichlorofluorescin diacetate (H_2_DCF-DA, ThermoFisher Scientific; Catalog number #D399), as described previously [141,142]. Incubation for both fluorescent probes, as well as washing and imaging, were conducted in a Krebs-Ringer solution (NaCl 115 mM, KCl 5 mM, NaHCO_3_ 10 mM, MgCl_2_ 2.5 mM, CaCl_2_ 2.5 mM, HEPES 20 mM) supplemented with 10 mM glucose. After 24 h of treatment with 10% patients’ serum, HUVECs were incubated with 2.5 μg/mL Hoechst 33342, trihydrochloride, trihydrate (ThermoFisher Scientific; Catalog number #H21492) for 30 min, in the dark, at room temperature. Then, HUVECs were washed and incubated with 10 μM H_2_DCF-DA for another 15 min and then washed 3 more times and incubated without any fluorescent probes for another 15 min, in the dark, at room temperature. Immediately after this step, cells were imaged using a Nikon CSU-W1 Spinning Disk confocal microscope, using a 40x objective (Nikon Corporation, Minato City, Tokyo, Japan). The non-fluorescent dye H2DCF-DA is a chemically reduced form of fluorescein and is cell-permeable; once intracellular esterases cleave off the diacetate (DA) moiety, H_2_DCF becomes sensitive to oxidation by ROS: in its oxidized form, dichlorofluorescein (DCF) is highly fluorescent and easily detectable. Thus, cells were excited with a laser at wavelengths 405 nm and 488 nm for Hoechst and DCF, respectively. Light emission was detected using 455/50 and 520/40 filters for Hoechst and DCF, respectively. The same settings (including laser intensity, exposure time, and pinhole width) were used for imaging in all experimental groups. In order to prevent DCF photodynamic reactions, fields of view search and focusing were performed using a Hoechst signal. Quantification of DCF fluorescence was performed using ImageJ software.

Mitochondrial ROS production was quantified by MitoSOX™ Red (ThermoFisher Scientific, #M36008) as previously described and validated [43,140,142,143,144]. Briefly, after 24 h of treatment with 10% serum, HUVECs were incubated with 200 nM MitoTracker™ Green FM (MTG, ThermoFisher Scientific; Catalog number #M7514) in EGM-2 medium in the dark for 1 h at 37 °C and 5% CO_2_. Then, the cells were washed with EGM-2 medium and incubated with 5 µM MitoSOX™ Red in EGM-2 medium in the dark for 10 min at 37 °C and 5% CO_2_. After incubation, cells were washed with EGM-2 medium and imaged at the Nikon CSU-W1 Spinning Disk confocal microscope using a 100× objective (Nikon Corporation). Cells were excited with a laser at wavelengths 488 nm and 561 nm for MTG and MitoSOX, respectively. Light emission was detected using 520/40 and 605/52 filters for MTG and MitoSOX, respectively. The same settings (including laser intensity, exposure time, and pinhole width) were used for imaging all experimental groups. In order to prevent MitoSOX photodynamic reaction, fields of view search and focusing were performed using a MTG signal. MitoSOX fluorescence intensity was quantified using NIS-Elements software (Nikon Corporation).

MDA and 4-HNE levels were measured by using commercially available kits (both from Abcam, Cambridge, UK), following the manufacturer’s instructions [57]. We quantified lipid peroxidation using C_11_-BODIPY^581/591^ (ThermoFisher Scientific) as previously described [59,145,146]. Briefly, HUVECs were incubated with 2 μM C_11_-BODIPY^581/591^ for 30 minutes (37 °C in the dark) and then fluorometrically measured (with an excitation wavelength of 581 nm and an emission wavelength of 591 nm) using a microplate reader.

### 4.3. Immunoblotting and RT-qPCR

Immunoblotting assays were performed as previously described by our group [144,147]; the intensity of the bands was quantified using the open-source image processing package FIJI (Fiji is Just ImageJ). The antibody for GPX4 was purchased from GeneTex (Irvine, CA, USA; Catalog number: #GT1282); the antibody for SCL7A11 was purchased from Cell Signaling (Danvers, MA, USA; Catalog number: #12691); the antibody for FTH1 was purchased from ABclonal Technology (Woburn, MA, USA; Catalog number: #A19544); the antibody for SAT1 was purchased from Cell Signaling (Catalog number: #61586); the antibody for GAPDH was purchased from Novus Biologicals (Centennial, CO, USA; Catalog number: #NB300-221). RT-qPCR was performed using a SYBR Green mix as we previously described and validated [148,149,150] using GAPDH as an internal standard; primer sequences are listed in Table 1.

### 4.4. Statistical Analysis

Statistical analyses were performed using GraphPad 9 (Dotmatics, Boston, MA, USA). Statistical significance, set at *p* < 0.05, was tested using the two-way ANOVA followed by Tukey–Kramer multiple comparison test or the non-parametric Mann–Whitney U test, as appropriate. 

## 5. Conclusions

Taken together, our results indicate that the serum from COVID-19 patients who did not survive induces oxidative stress, lipid peroxidation, and ferroptosis in human endothelial cells through a mechanism that depends on TNF-α.

## Figures and Tables

**Figure 1 antioxidants-12-00326-f001:**
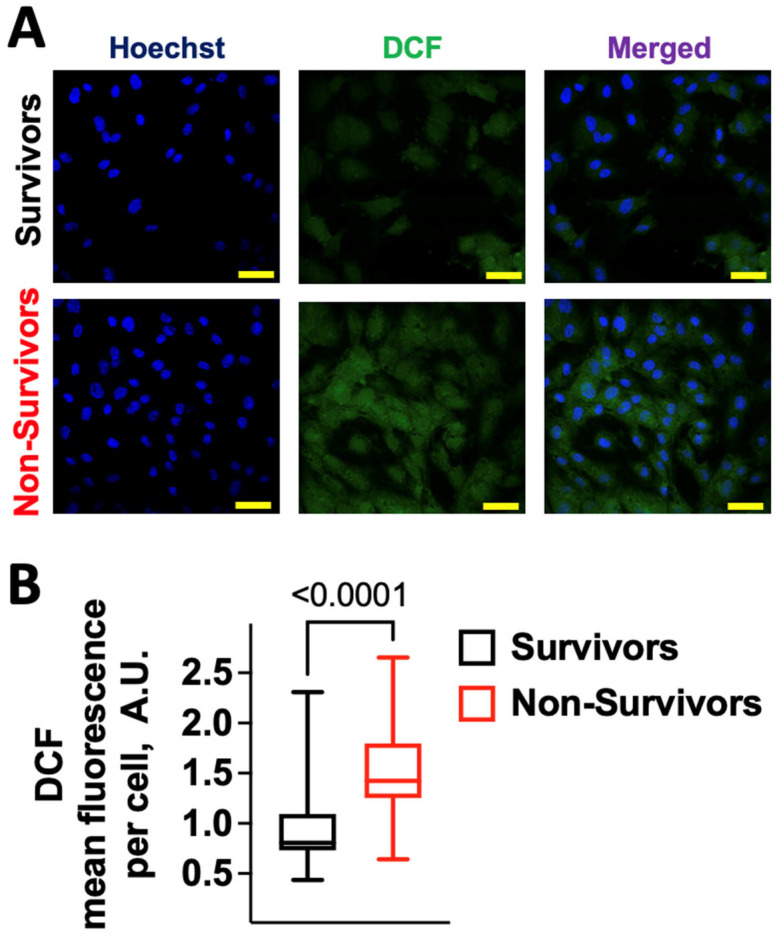
Serum from patients who did not survive COVID-19 promotes oxidative stress in human endothelial cells. (**A**) Representative microphotographs of DCF and Hoechst staining of HUVECs treated for 24 h with 10% serum obtained on hospital admission from patients who survived or succumbed to COVID-19; scale bar: 50 μm. (**B**) Quantification of DCF fluorescence intensity. Data were obtained from at least three independent experiments and are presented as a box-and-whiskers plot showing the median and the 5th–95th percentiles; A.U.: arbitrary units.

**Figure 2 antioxidants-12-00326-f002:**
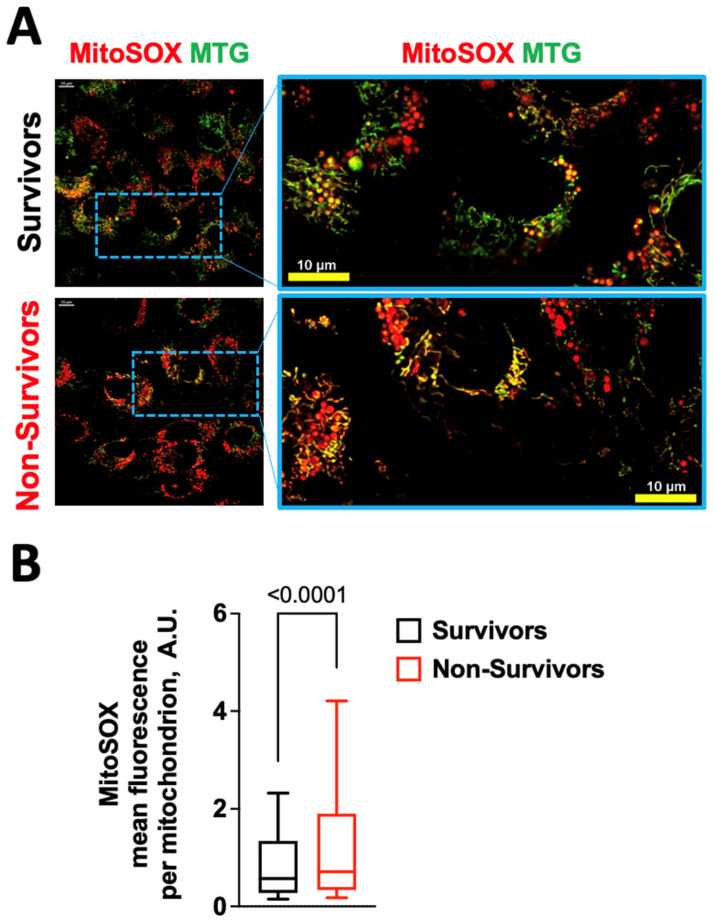
Serum from patients who did not survive COVID-19 increases mitochondrial ROS production. (**A**) Representative microphotographs of MitoSOX Red and MitoTracker Green FM (MTG) staining of HUVECs treated for 24 h with 10% serum from patients who survived or did not survive COVID-19. (**B**) Quantification of MitoSOX Red fluorescence intensity. Data are from at least three independent experiments and are presented as a box-and-whiskers plot showing the median and the 5th–95th percentiles; A.U.: arbitrary units.

**Figure 3 antioxidants-12-00326-f003:**
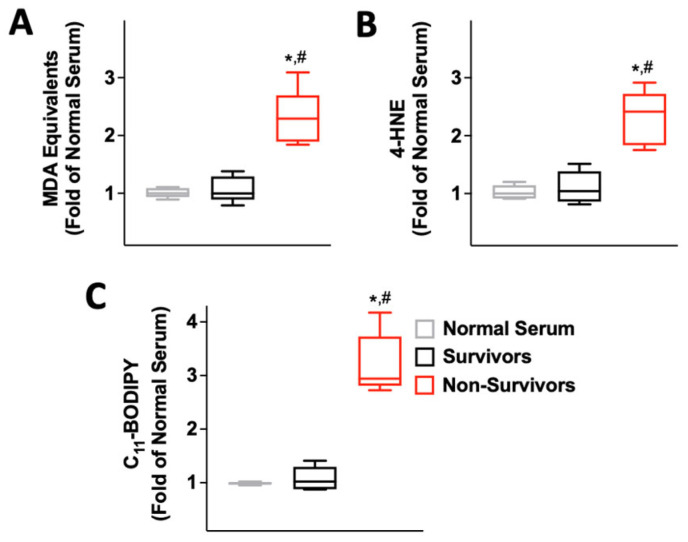
Lipid peroxidation in HUVECs is increased by serum from patients who did not survive COVID-19. Three different assays, namely the quantification of MDA equivalents (**A**), 4HNE (**B**), and C_11_-BODIPY (**C**), confirmed that the incubation of HUVECs for 24 h with serum from COVID-19 patients who did not survive significantly increased lipid peroxidation. All experiments were performed in triplicate; the box-and-whiskers graphs show the median and the 5th–95th percentiles; *: *p* < 0.01 vs. Normal Serum, #: *p* < 0.01 vs. Survivors.

**Figure 4 antioxidants-12-00326-f004:**
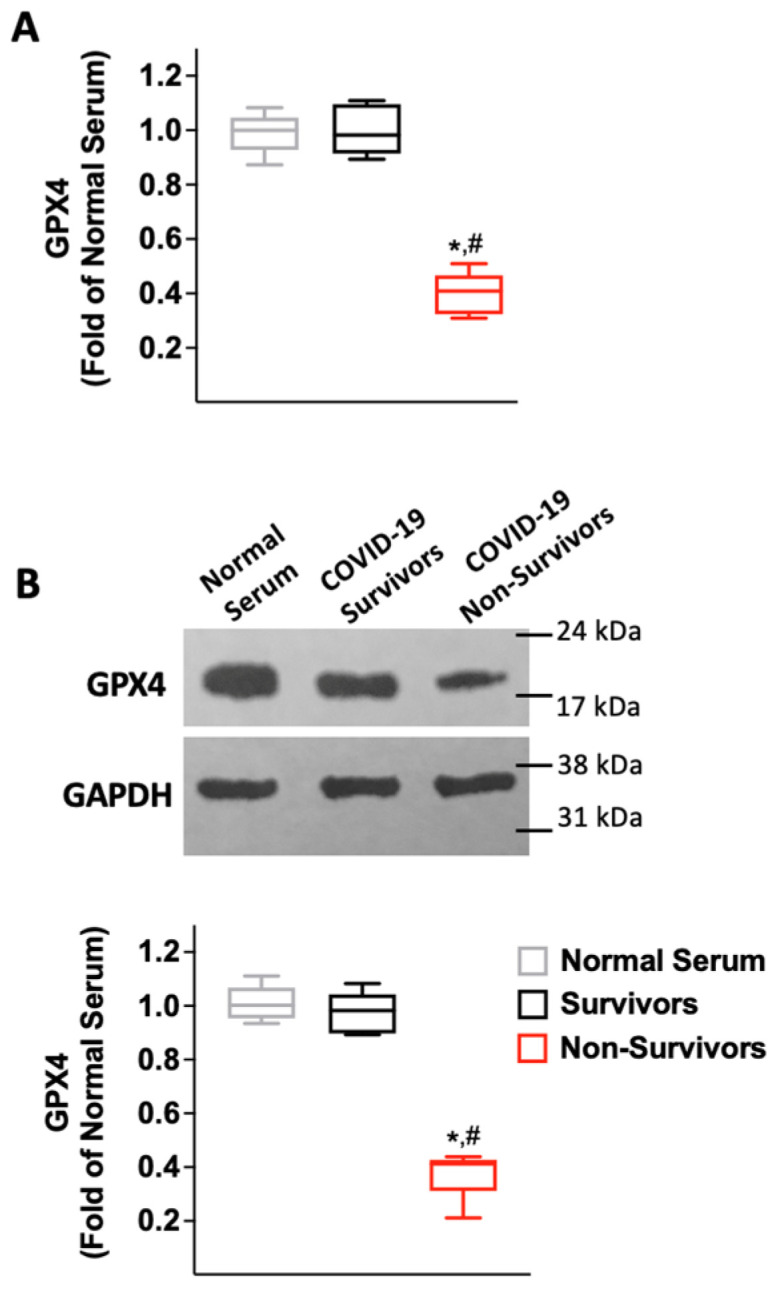
GPX4 expression in human endothelial cells is reduced by serum from patients who did not survive COVID-19. GPX4 levels were measured by RT-qPCR (**A**) and by immunoblot (**B**) in HUVECs, normalizing to glyceraldehyde 3-phosphate dehydrogenase (GAPDH); panel B shows representative blots from triplicate experiments (top) and their quantification (bottom). All experiments were performed at least in triplicate; the box-and-whiskers graphs show the median and the 5th–95th percentiles; *: *p* < 0.01 vs. Normal Serum, #: *p* < 0.01 vs. COVID-19 Survivors. Sequences of oligonucleotide primers are reported in Table 1.

**Figure 5 antioxidants-12-00326-f005:**
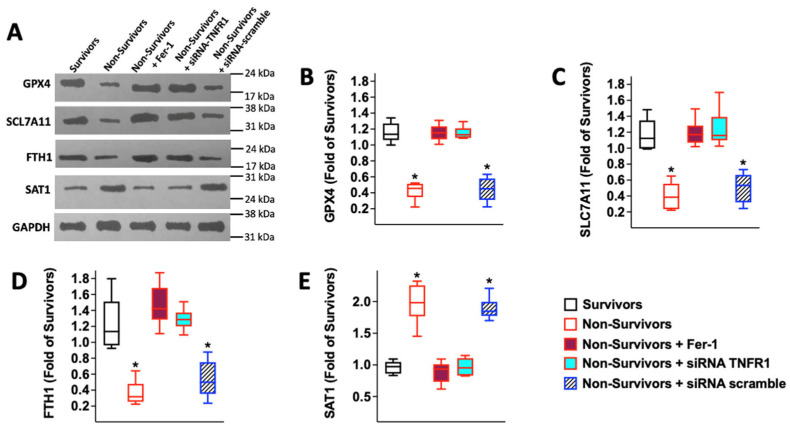
COVID-19 serum from non-survivors regulates ferroptosis via TNFR1 in HUVECs. Representative immunoblots are shown in panel (**A**), quantified in panel (**B**) (GPX4), (**C**) (SCL7A11), (**D**) (FTH1), and (**E**) (SAT1). All experiments were performed at least in triplicate; the box-and-whiskers graphs show the median and the 5th–95th percentiles; *: *p* < 0.01 vs. Survivors.

**Table 1 antioxidants-12-00326-t001:** Oligonucleotide sequences of the primers used for RT-qPCR.

	Primer	Sequence (5′-3′)	Amplicon (bp)
**GPX4**	** *Forward* **	GAG ATC AAA GAG TTC GCC GC	102
	** *Reverse* **	CTT CAT CCA CTT CCA CAG CG	
**TNFR1**	** *Forward* **	TTG TAT GGC CCC AAC TGT CT	99
	** *Reverse* **	CTG GCT CAA GTC CTT CCT CA	
**GAPDH**	** *Forward* **	GGC TCC CTT GGG TAT ATG GT	94
	** *Reverse* **	TTG ATT TTG GAG GGA TCT CG	

## Data Availability

Data supporting reported results are contained within this article and its Appendix A.

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
