# Peer review of "COVID-19 Causes Ferroptosis and Oxidative Stress in Human Endothelial Cells"

_antioxidants, 2023, doi:10.3390/antiox12020326_

Round 1

Reviewer 1 Report

The manuscript from Jankauskas and co-workers shows that sera obtained from COVID-19 patients which died due to the infection significantly increase lipoperoxidation in cultured HUVECs, differently from serum samples obtained from survivors of the SARS-CoV-2 infection.

Despite predictable, the results are still relevant, the methodology employed is appropriate and the manuscript is well written.

Minor points:

1. Under methods, it should be corrected that the species detected by fluorescence emission at 488 nm is DCF (and not H2DCF-DA), as reduced H2DCF-DA is the reagent that firstly undergoes deacetylation by intracellular esterase action, and H2DCF is further oxidized to DCF by several ROS.

2. Regarding MDA quantification, the method used is based on the reaction of aldehydes, generated as lipoperoxidation by-products, with thiobarbituric acid. Despite the kit uses MDA as the standard calibrator, the signal obtained from the HUVEC samples is not solely due to this aldehyde. In this way, the results must not be expressed as "MDA" but rather as "MDA equivalents".

3. The authors measured TNF-alpha concentrations in the sera obtained from the different groups of patients and refer that these results are shown in the supplementary Table S1. However, this material is not available (at least, to me; I have only found one supplementary pdf file showing the WB images). Considering the relevance of directly showing these TNF concentrations to the readers in the main manuscript, I strongly suggest that these results (and their statistical analysis) should be embedded in the body text (item 2.5).

Author Response

We thank this Reviewer for the kind words of appreciation.

All the minor points have been rectified, as recommended by this Reviewer; thanks.

Reviewer 2 Report

The paper by Jankauskas et al reports the effect of serum obtained from COVID-19 patients on HUVECs. Interestingly, they found that those affected by a very aggressive form (fatal outcome) induced a more severe in vitro endothelial damage, particulrarly represented by oxidative stress and lipid peroxidation. They also address the mechanism and find that TNF-aplpha pathway (based on siRNA experiments) and ferroptosis (based on ferrostatin-1 exp) are likely involved. 

Overall, the paper is sound and informative, and the experimental  flow is well conceived. 

Several minor concerns however have been raised in a collaborative spirit to enhance the impact of the manuscript.

In the Introduction, the general role of oxidative stress on endothelium could be anticipated, to help the reader to focus on the subject.

A table reporting the main features of patients (survivors vs non-survivors) should be reported). 

In figs 1 and 2, the control of "normal serum" is lacking. Please add it or at least explain the rationale why it should not be reported. 

Similarly, fig 5 also lacks controls. For instance,"normal serum", but also a vehicle for Fer-1, and an expression control for TNFR1 in siRNA experiments. 

Moreover, as proof of concept, a stimulation with TNF-alpha on HUVEC would add a great deal of soundness.

Author Response

R: We thank this Reviewer for her/his kind words of appreciation towards our work.

R: We thank this Reviewer for this pertinent remark; in the revised version of our paper we have expanded the Introduction to discuss the functional role of oxidative stress on endothelium.

R: We thank this Reviewer for these suggestions. Table S1 is reporting the main features of patients (survivors vs non-survivors). We added a new Figure (Supplementary Figure S!) confirming the efficiency of TNFR1 silencing, as requested. We also agree with this Reviewer that showing the effects normal serum could have strengthened our results; however, as stated in the introduction, the goal of this research project was to test the hypothesis that oxidative stress and lipid peroxidation induced by COVID-19 in endothelial cells could be linked to the disease outcome; we respectfully believe that showing (or not) any difference in the phenotype triggered by normal serum vs COVID-19 serum in all the experimental settings would have not changed our conclusions. Concerning the experiments with TNFa, we thank this Reviewer for this insightful suggestion: we are currently working on a dedicated project (not dealing with COVID-19, to actually dissect the mechanistic role of TNFa signaling pathway in ferroptosis and oxytosis). We have expanded the section addressing the limitations of our study to acknowledge these aspects.

Round 2

Reviewer 2 Report

The authors have satisfactorily addressed all previous concerns.